# Possible Epigenetic Origin of a Recurrent Gynandromorph Pattern in *Megachile* Wild Bees

**DOI:** 10.3390/insects12050437

**Published:** 2021-05-12

**Authors:** Daniele Sommaggio, Giuseppe Fusco, Marco Uliana, Alessandro Minelli

**Affiliations:** 1Dipartimento di Scienze e Tecnologie Agro-Alimentari (DISTAL), Alma Mater Studiorum-Università di Bologna, Viale G. Fanin 42, 40127 Bologna, BO, Italy; daniele.sommaggio@unipd.it; 2Department of Biology, University of Padova, Via U. Bassi 58/B, 35131 Padova, PD, Italy; alessandro.minelli@unipd.it; 3Via Cavaizza 16/B, 35020 Codevigo, PD, Italy; marco.uliana.1@gmail.com

**Keywords:** Hymenoptera, sex determination, sex differentiation, genetic mosaic, epigenetic sex mosaic, alternative splicing, developmental stability, modularity

## Abstract

**Simple Summary:**

Gynandromorphs, i.e., individuals with a mix of male and female body parts, are known for many species of insects and other animals with separate sexes. This anomaly is generally regarded as the result of localized genetic mutations in sex-determining genes. We analyzed the specific mix of male and female characters in naturally occurring gynandromorphs of 21 species of the wild bee genus *Megachile* and found a recurrent pattern. Based on the regularity of this pattern, and the current knowledge on sex determination and sex differentiation in the relatively closely-related honey bee, we argue that the origin of these composite phenotypes is possibly epigenetic, rather than genetic, i.e., produced by some defects in the maintenance of the regulatory signals that control sex differentiation at the level of single cell lineages, rather than triggered by genetic mutations.

**Abstract:**

Gynandromorphs, i.e., individuals with a mix of male and female traits, are common in the wild bees of the genus *Megachile* (Hymenoptera, Apoidea). We described new transverse gynandromorphs in *Megachile pilidens* Alfkeen, 1924 and analyze the spatial distribution of body parts with male vs. female phenotype hitherto recorded in the transverse gynandromorphs of the genus *Megachile*. We identified 10 different arrangements, nine of which are minor variants of a very general pattern, with a combination of male and female traits largely shared by the gynandromorphs recorded in 20 out of 21 *Megachile* species in our dataset. Based on the recurrence of the same gynandromorph pattern, the current knowledge on sex determination and sex differentiation in the honey bee, and the results of recent gene-knockdown experiments in these insects, we suggest that these composite phenotypes are possibly epigenetic, rather than genetic, mosaics, with individual body parts of either male or female phenotype according to the locally expressed product of the alternative splicing of sex-determining gene transcripts.

## 1. Introduction

In separate-sex animals, abnormal individuals with a mix of phenotypically male and phenotypically female anatomical parts are known as gynandromorphs. These are usually described as genetic or chimeric mosaics that consist of both male and female cells (e.g., [1,2]). Gynandromorphism can occur when sexual differentiation is not regulated at a systemic level and, depending on the species’ developmental cell lineages and the stage of development in which the anomaly occurs, the distribution of “male tissues” and “female tissues” can exhibit some kind of symmetry (e.g., bilateral) or produce a patchwork throughout the body. Many cases have been described, mostly among insects, chelicerates, decapod crustaceans, and birds [3].

In the literature, the terms gynandromorph and intersex have been used for individuals with a mix of male and female features, either as synonyms or to distinguish different classes of sex abnormalities, the latter either in terms of morphological patterns or of putative causes. We will discuss this issue in general terms and from a historical perspective in a separate paper [4]. Here, we adopt the terms gynandromorph for description and discussion of old and new cases in the wild bees of the genus *Megachile*, with the simple meaning of a morphological mix of male and female traits, in the virtual absence of parts with intermediate sexual phenotype and with no commitment to a specific causal explanation of the abnormalities.

Gynandromorphism is well known in bees (Hymenoptera, Apoidea, Anthophila; since Sphecoidea proved to be a paraphyletic taxon, the name Apoidea has been used to include both solitary wasps and bees, whereas Anthophila is used to include the monophyletic set of bee families [5,6]). Excluding the extensive literature about gynandromorphism in *Apis mellifera* Linnaeus, 1758, 134 cases have been recorded in all bee families except the Australian, species-poor Stenotritidae [7,8,9]. Gynandromorphism seems to be particularly common in Megachilidae, which alone accounts for more than one-third of all cases. In the huge genus *Megachile* Latreille, 1802 (more than 1400 species in 55 subgenera described thus far [10,11,12]) sexual dimorphism is conspicuous, which could contribute to explaining the high number of gynadromorphs reported.

The discovery in the wild bee *Megachile pilidens* Alfkeen, 1924 of several gynandromorphs, described below, has prompted a re-visitation of previous interpretations of gynandromorphs in a diversity of *Megachile* species, and suggests a new interpretation in the light of the current knowledge about sex differentiation in the honeybee.

Different classifications of gynandromorph hymenopterans have been suggested (e.g., [7,13,14,15]). Here we follow Michez et al. [7], who classified bee gynandromorphs into three categories: (i) transverse, when “sex characters are distributed in two asymmetrical parts” [7] (p. 367), (ii) bilateral, when one side of the body has male characters and the other female; (iii) mosaic, when male and female characters are distributed randomly in the bee. To better circumscribe the class of gynandromorphs discussed in the present paper, we qualify a gynandromorph as transverse when male and female traits coexist in the same individual, but each trait is present in the same state (either male or female) on both sides of the body. For example, a specimen with both antennae, mandibles, and front legs showing male characters, while hind legs and metasoma segments are typically female, is a transverse gynandromorph. In this use, transverse gynandromorphs is the same as the “frontal type” in [13,14]. We focused on transverse gynandromorphs in *Megachile*, because in this genus bilateral specimens have been recorded only in two species (*M. latimanus* Say, 1823 and *M. willughbiella* (Kirby, 1802)) and a mosaic specimen (actually, a partial bilateral gynandromorph) only in one species (*M. rotundata* Smith, 1853) [16,17,18,19,20].

We argue that these transverse gynandromorphs might not be caused by genetic differences between regional or local clusters of cells, but by differences in the regulation of gene expression affecting cell differentiation. As discussed below, rather than nuclear or chromosomal mosaics, *Megachile* gynandromorphs are possibly *epigenetic sex mosaics*, with body parts of either male or female phenotype according to the locally expressed product of alternative splicing of sex-determining gene transcripts.

## 2. Materials and Methods

### 2.1. The Dataset

Male or female character state was recorded for all dimorphic characters for which there is adequate information in the literature (Figure 1; see Appendix A for the complete list of gynandromorphs considered). In particular, we focused on the following ten characters (character names largely after [10,21]):

**Number of antennal segments** (Asn): In Anthophila, the number of flagellomeres is 10 in females and 11 in males. In some gynandromorphs (e.g., *M. vidua* and *M. parallela* [14]) the number of flagellomeres is 10 but the terminal one is incompletely divided on one side of the antenna. In these cases, which we interpreted as an incomplete fusion of two segments or an incomplete division of the last segment, we recorded the antenna as 11-jointed (male).

**Length of antennal segments** (Asl): In *Megachile*, the length/width ratio of antennal flagellomeres is sexually dimorphic; male flagellomeres are elongated, and distinctly longer than wide.

**Face pubescence** (Fap): In *Megachile*, hairs on the front are well developed in both sexes, but on the face below the antenna and on clypeus, hairs are long and dense in male, sparse in female. Fap state was not described in the *M. detersa* gynandromorph [22]. In *M. curvipes*, pilosity “is dense enough to hide the surface, but is not quite of the male type. The clypeus is thus about intermediate between the typical male and female condition” [16] (p.67). In both cases (*M. detersa* and *M. curvipes*) we scored the character state as missing data.

**Mandibles** (Man): Mandibles are well developed in *Megachile* females, where several species use them to cut leaves to build a nest for eggs and larvae. Despite conspicuous variation in mandible size and shape in the genus (mandible shape is often diagnostic for subgenera [10]), female mandibles have an expanded cutting edge, with three to five teeth, whereas male mandibles are not expanded, and usually have three large teeth. Man state was not described in the *M. detersa* gynandromorph [22]. In *M. curvipes*, mandibles are similar to the female type, except for the shape of the teeth [13]. In both cases (*M. detersa* and *M. curvipes*), we scored the character state as missing data.

**Front tarsi and/or femora** (Ftf): Male front tarsi, especially the metatarsi, are usually broadly dilated in the male, whereas female ones are slender. In many species, front tarsi are more conspicuously colored in male than in female. In few cases, e.g., in *M.* (*Eutricharaea*) *pilidens*, the male front femora, rather than front tarsi, are dilated. In few subgenera, e.g., *Litomegachile* and *Neochelynia*, front legs are female-like and not sexually dimorphic. Accordingly, in *M.* (*Litomegachile*) *onobrychidis* and *M.* (*Neochelynia*) *uniformis* we scored the character state as missing data. 

**Front coxae** (Cox): In several *Megachile* subgenera, males have distinctive spines on the coxae, lacking in females. These spines may be reduced and even completely lacking in the males of a few subgenera, such as *Neochelynia* and *Ptilosaurus*. Gynandromorphs reported in the literature mostly belong to species with well-developed spines, with the exception of *Megachile* (*Ptilosaurus*) *bertonii* and *M.* (*Neochelynia*) *uniformis*; in these cases, we scored the character state as missing data.

**Scopa** (Sco): The scopa is the pollen-collecting apparatus, and in *Megachile* it is found on the ventral surface of the metasoma. Female sternites from the second backward are covered with a dense brush of long and stiff hairs. Sternites of male metasoma are usually pubescent, but hairs are shorter and thinner than in females, and not functional to carry pollen. In gynandromorphs, the phenotype is usually female (scopa present) or male (scopa absent); however, we scored it as male also in the cases of *M.* (*Chelostomoides*) *angelorum*, where the hairs on metasoma sternites are longer than in male but nonetheless non-functional.

**Metasoma segments** (Mts): In *Megachile*, there are six tergites and six sternites in the female metasoma, whereas in males there are six tergites, but only four (rarely three) visible sternites.

**Sixth tergite of the metasoma** (Tg6): In males, the sixth tergite of the metasoma bends ventrally and presents a transverse carina at the apical end, in several species armed with spines. In females the sixth tergite is flatter and no carina is present.

**External genitalia** (Gen): Usually the female sting apparatus (stylus and lancet) projects from the apex of the metasoma.

Other dimorphic characters, for example, the gena (wider in males than in females), the claws (usually cleft in males, simple in females), and the hind metatarsi (simple in males, dilated in females, with denser hairs) are considered in the Discussion; however, we did not include them in the dataset because in several species they are not sexually dimorphic and/or the state of the character is unknown for several gynandromorphs described in the literature.

With the term “pattern” we will refer to any distinctive mix of male and female character states for the ten characters included in the dataset.

### 2.2. New Gynandromorphs in Megachile pilidens

In the next section we describe a series of gynandromorphs discovered in *Megachile* (*Eutricharaea*) *pilidens*, a species from which the phenomenon has been previously reported [23], but not described in detail. Six gynandromorph specimens were collected at two localities of the Berici Hills (North-Eastern Italy) with Malaise traps (detailed information on locality habitats in [24,25]); they are preserved in the collection of one of the authors (DS).

In the same localities, during the same sampling effort with Malaise traps, 27 males and 10 females were also collected. Thus, gynandromorphs represent 13.9% of the collected specimens, a high percentage with respect to other reports, in which they are usually considered to be rare, e.g., [15].

## 3. Results

### 3.1. Description of the New M. pilidens Gynandromorphs

The six *M. pilidens* gynandromorphs all present the same pattern (Figure 2), with the following male traits:

**Number of antennal segments**: The number of flagellomeres is 11, except for one specimen with 10, but the last one is almost twice the length of the previous one (Figure 3). In two specimens the number of flagellomeres is 11, but the last two are partially fused.

**Length of antennal segments**: Flagellomeres distinctly longer than wide (Figure 3).

**Face pubescence**: Dense hairs on the whole face, including clypeus (Figure 3).

**Mandibles**: Three large teeth, the proximal one very large, plus an additional ventral tooth (absent in females) (Figure 3).

**Front femora**: Larger than in females; ventral apical margin enlarged and yellow (completely black in females) (Figure 4).

**Front coxae**: Spine present (Figure 4).

**Scopa**: Sternites completely bare in the middle, a few hairs present on posterior and lateral margin (Figure 5).

**Genae:** Area between eyes and occipital groove larger than in females.

**Claws**: With a subapical tooth, similar in length to the main one (claws simple in females) (Figure 6).

**Hind metatarsi**: Simple and with long white hairs (dilated and with dense yellow pubescence in females) (Figure 6).

Our *M. pilidens* gynandromorphs have the following female traits:

**Metasomal segments**: Six sternites (Figure 5).

**Sixth metasomal tergite**: Simple, without a carina (Figure 7).

**External genitals**: Sting apparatus present.

**Mesoscutum and scutellum**: Both sclerites well developed, scutellum slightly larger than wide.

### 3.2. Transverse Gynandromorph Patterns in Megachile

Transverse gynandromorphs have been recorded thus far in 21 *Megachile* species. In two species (*M. angelarum* and *M. vidua*), two different gynandromorph patterns have been recorded, for a total of 23 species patterns and 10 distinguishable patterns in the dataset (Figure 1). Pattern 1, found in *M maritima* [26], is the more deviant pattern with respect to all others, almost opposite to the very frequent pattern 10, and the only one with male external genitalia. Here we report additional information available for gynandromorphs exhibiting some of the other patterns.

Pattern 4. Recorded only in a specimen of *M. montezuma* [27] with additional female head features (interocellar distance, ocelli–occiput distance and genae).

Pattern 5. In *M. deceptoria* and *M. picicornis* gynandromorphs, claws are bifurcate [9]. In *M. otomita*, the spine on the front coxa and the front femora are enlarged, although less than in normal males; the pubescence on the tergites is whitish, while it is reddish to yellowish in normal specimens of both sexes [8]. In the *M. vidua* gynandromorphs, the antennal phenotype is variable: there are 10 flagellomeres, but with some differences in the size of the terminal flagellomere between specimens, and between left and right antenna in one of them [14]. In *M. perihirta*, the phenotype of the front legs is male proximally, female distally: front femora colored as in male, tibiae entirely dark as in female; basal half of front tarsi enlarged and whitish as in male, but apical tarsi narrowed and black as in female; middle legs similar to those in males (enlarged and with “keel-like protuberance” in ventral part of metatarsi, even if less developed than in males); hind legs similar to those in females, with large metatarsi [14].

Pattern 6. In a *M. gathela* gynandromorph [28], the only occurrence of this pattern, the head has several female characters (clypeus punctuation, ocelli–occiput distance, genae width) and the claws are bifurcate. In this species, metasoma sternites are sexually dimorphic and the gynandromorph presents a complex combination of male and female features: tergites 1–6 are described as male (even if the carina of tergite 6 projects more ventrally than in males), hairs on sternites 1–3 are similar to male ones (“soft, simple and comparatively elongated”, not functional to collect pollen), the apical margin of sternite 4 has hairs similar to those found in female scopa and sternites 5–6 are exposed, and the fifth with a scopa as in the female.

Pattern 7. Recorded in 15 specimens of *M. detersa*. Unfortunately, the description is incomplete, and information is missing for several characters in our dataset.

Pattern 8. In *M. rubricata*, the gynandromorph head mostly exhibits male character states (also for ocelli–occiput distance), except for the mandibles, and typically male veins have been reported in the fore wings [29].

Pattern 8 or 9. Mitchell [14] described several gynandromorphs of *M. intergradus* which he divided into three categories; however not all categories are well described and/or differences are not clear, for this reason, we prefer to include in our analysis only the third one (“Sex intergrade 3”). However, because mandibles were not described, this record could belong to either pattern 8 or 9. In *M. intergradus* gynandromorphs claws are bifurcate as in males.

Pattern 10. This is the more frequent pattern, with records belonging to nine species, including the *M. pilidens* specimens described in the present paper. Claws are bifurcate as in males in *M. bertoni* [14]. In *M. uniformis* [15] the metasoma is “ferruginous and punctate” as in the male. The only available description of the behavior of a *Megachile* gynandromorph was given for a *M. gemula* individual that tried to copulate with a female, despite the absence of male genitalia [14].

### 3.3. Frequency Analysis

Frequency analysis is based on 10 characters which can alternatively occur in male or female state in two different datasets: (i) the *species-pattern dataset* coincides with the source dataset and contains 23 records, one for each species, except for two records for each of the two species in which two alternative patterns were recorded; (ii) the *pattern dataset* derives from the former and contains 10 records corresponding to the 10 alternative patterns observed in one or more species each.

Most species patterns, in addition to most patterns, show a combination of opposite states for two disjoint sets of characters (Figure 8). Male state in characters Asl, Fap, Man, Ftf, and Cox (anterior set) is frequently associated with the female state in characters Mts, Tg6 and Gen (posterior set). The behavior of the two remaining characters, Asn and Sco, is less definite. Character-state Asn^F^ combines with the male state of the anterior-set characters in about 35% of species patterns and 45% of patterns, whereas character-state Asn^M^ combines with female state of the posterior-set characters in 55% of species patterns and 40% of patterns. Character-state Sco^F^ combines with the male state of the anterior-set characters in about 20% of species patterns and 35% of patterns, whereas character-state Sco^M^ combines with female state of the posterior-set characters in 75% of species patterns and 45% of patterns.

Overall, there is a strong concordance in the sex state within each of two sets of characters and a strong discordance between the two sets (Figure 9). In the species-pattern dataset, characters Asn and Sco tend to be more strongly associated with the anterior character set (ca. 60% and 75%, respectively), whereas in the pattern dataset, the two characters are nearly indifferently associated with one or the other of the two character sets.

## 4. Discussion

### 4.1. New Megachile Gynandromorphs

The remarkable number of gynandromorphs found in Berici Hills (13.9% of all specimens of *Megachile pilidens* collected there) raises an interesting question about population dynamics. If gynandromorphs behave like sexually mature males, as suggested by the only record of mating behavior of a *Megachile* gynandromorph available thus far [14], a high percentage of specimens would not leave offspring. If gynandromorphs behave like females, they would not be able to produce a nest with pollen as food for the larvae (recall that the male-like mandibles of gynandromorphs do not allow them to cut leaves for nest building, and they lack a scopa to collect pollen). At best, they might deposit eggs in nests of functional females, thus behaving as cleptoparasites; see [9] about the intriguing similarity between *Megachile* gynandromorphs and cleptoparasite females of the genus *Coelioxys*, the sister group of *Megachile*.

### 4.2. Topographic Considerations

Most of the states of the examined characters in *Megachile* gynandromorphs correspond to the phenotype observed either in normal males or normal females.

A recurrent, general pattern emerges from frequency analysis, with an anterior set of characters with male state (Asl, Fap, Man, Ftf, and Cox), a posterior set of characters with female state (Mts, Tg6 and Gen), and two characters, one in the anterior of the body (Asn) and one in the posterior of the body (Sco), that exhibit male or female state largely independent of the other characters and of each other. Only the gynandromorph of *M. maritima* presents a pattern wholly opposite to the general one.

Topographically, the boundary between the two sets of characters passes transversally somewhere between the posterior end of the mesosoma and the anterior end of the metasoma. However, it must be noted that characters in the anterior set mostly involve features of the appendages, and are thus derived from the imaginal discs, whereas the characters in the posterior set mostly involve features of the body trunk.

To our knowledge, no information is available on the internal anatomy of *Megachile* gynandromorphs, and too little is known on the metamorphosis in these insects to speculate about this question.

Adopting a finer-grained level of description, some traits of some gynandromorph specimens are not completely masculine or feminine. Antennae and legs may show a polarity of phenotypic conditions, distinctly male in the proximal part of the appendix, but with a transition to (or overlapping with) female elements in the distal part. This polarity is manifested in the fusion/separation of the last antennal flagellomeres in *M. vidua* and *M. parallela* [14], whereas in the legs of *M. perihirta* [14] the coxal spine is present as in the male, the femurs are colored as in the male, and the tarsi are colored and expanded as in the male, except in the apical part where they are narrow and black as in the female. On a different body axis, the ventral sclerites of the metasoma in the *M. gathela* gynandromorph change from male to female from the anterior to posterior [28]. This similarity in sex polarity between proximo-distal and antero-posterior axes (although recorded here in different individuals) is suggestive of the principle of paramorphism, according to which there would be strict correspondence between the developmental control along the axes of the appendages and the main body axis of the same animal [30].

A final note regards the behavior of Asn and Sco, which are not strictly associated with either the anterior or the posterior set of characters. Despite this similarity, the causes of their relative independence of other characters could be different. We only note that both are expressed in tissues (antenna and ventral metasoma, respectively), together with another sexually dimorphic character (Asl and Mts, respectively) from which they can differ in sex state, raising the possibility of differently controlled overlapping processes of sex differentiation.

### 4.3. Developmental Interpretation

With the exception of *M. maritima* (see below), all *Megachile* gynandromorphs show a general pattern of combined male and female parts. We attempt to read these cases in the light of what is known about sex differentiation in the honeybee, which under many aspects is probably generalizable to most of the Hymenoptera.

#### 4.3.1. Sex Determination and Sexual Differentiation in the Honeybee

The sex-determination system in the honeybee, and in Hymenoptera in general, is haplodiploid: fertilized eggs develop into diploid individuals of female sex, as a rule, whereas unfertilized eggs develop parthenogenetically into haploid males. Diploid male, however, have been occasionally recorded in a number of hymenopteran species [31,32,33]; this eventually led to the discovery of the mechanism of complementary sex determination (CSD) [3]. Fertilized eggs that are heterozygous at one or more Sex Determination Loci (SDL) develop into females, whereas unfertilized hemizygous or fertilized homozygous ones differentiate into haploid or diploid males, respectively.

Complementary sex determination was first suggested by genetic studies in the parasitic wasp *Habrobracon hebetor* (Say, 1836) ([34,35], sub *Bracon hebetor*). This mechanism is known only in species belonging to four of the 21 hymenopteran superfamilies (Apoidea, Vespoidea, Ichneumonoidea, Tenthredinoidea); however, two of these (Vespoidea and Ichneumonoidea) also include species where this mechanism does not operate; non-CSD species are also known from another four superfamilies (Chalcidoidea, Cynipoidea, Proctotrupoidea, Chrysidoidea). No data are available for the remaining major lineages within the order [33]; however, the phylogenetic distribution of CSD among the Hymenoptera suggests that this is the ancestral mechanism for haplodiploidy in the group [36].

In the honeybee, complementary sex determination is controlled at a single locus [37,38,39,40], the *complementary sex determiner* (*csd*) gene. In this species, at least 15 allelic variants of *csd* have been discovered, which differ on average for ca. 3% of their amino acid residues [41,42]. Five amino acid differences and length variation between *csd* alleles in the potential specifying domain are sufficient to regularly induce femaleness [43].

Further downstream from the mechanism of sex determination in the honeybee, there is the activity of the *feminizer* (*fem*) gene, an orthologue of the *Drosophila* gene *transformer* (*tra*) [44,45,46]. *fem* is a paralogous version of *csd*, from which it split by duplication before the divergence of the Aculeata, ~120 MYA [47].

*csd* is required to initiate sex-specific differentiation, whereas the *fem* activity is required to maintain the female determined pathway throughout development [48]. Limited to a restricted time window in early development, in late blastoderm stage, ca. 25–35 h after egg deposition, *csd* controls the alternative splicing of *fem* pre-mRNA. The mature mRNA products are either the male-specific variant, which encodes a non-functional product, or the female-specific variant, which encodes a functional regulative protein (Fem^F^). Fem^F^ is required to mediate the splicing of *double sex* (*dsx*) gene pre-mRNA into the female-specific *dsx*^F^ mRNA. *dsx* transcripts encode sex-specific transcription factors that regulate the expression of numerous target genes involved in various aspects of sexual differentiation. *dsx* belongs to the Dmrt gene family, whose members are specifically expressed during the development of the gonads of almost all bilaterians, where they promote the differentiation of male-specific traits and repress those specific to the female [3].

Key to our argument below is that at the level of each cell, the female-determined state is maintained through a positive autoregulatory activity of the Fem^F^ protein, which directs the processing of female-specific *fem*^F^ mRNA, thus providing a steady source of a female-specific signal required for the differentiation of the female phenotype, including adult somatic and reproductive traits [46]. In the absence of this sex-specific signal, the default regulatory state leads to sex differentiation as a male.

#### 4.3.2. The Phylogenetic Background—Apis vs. Megachile

No information about sex determination in Megachilidae is available to date [49], but the bee species for which there is some evidence of CSD are not restricted to the Apidae (13 species belonging to the genera *Apis*, *Bombus*, *Euglossa*, *Melipona*, *Scaptotrigona*, and *Trigona*), but also include the Halictidae (*Augochlorella*, *Halictus*, and *Lasioglossum*) and Andrenidae (*Andrena*) [49]. The two latter families are more distantly related to the Apidae than the Megachilidae are [50].

Sex differentiation has been studied thus far only in the honeybee, but phylogenetic considerations suggest that we can reasonably attempt an interpretation of *Megachile* gynandromorphs based on the mechanism of sex differentiation as known in that species. Family Megachilidae is sister to Apidae, the two families together forming the clade of the long-tongued bees [10]: this has been confirmed by molecular phylogenies [50,51,52] and *Megachile* has been accordingly selected as an outgroup to polarize a phylogeny of the Apidae [53]. The last common ancestor of Apidae and Megachilidae can be dated to the Cretaceous, about 95 MYA [50].

#### 4.3.3. Tentative Interpretation of Megachile Gynandromorphs

In the past, gynandromorphs have been tentatively explained in terms of delayed (embryonic) fertilization [54], polyspermy [55], and chromosome elimination [56]. Early efforts to explain gynandromorphs in the honeybee according to these models [57] required adjustments to cope with the assumption that male bees (drones) and the body parts of gynandromorphs exhibiting male traits are necessarily haploid. The discovery of complementary sex determination has disclosed new possible scenarios for the origin of gynandromorph bees. This suggests that in diploid gynandromorphs, female parts are heterozygous at the *csd* locus, whereas male parts are either hemizygous (as in normal drones) or homozygous at the same locus [7]. The latter hypothesis would imply mutation and/or inhibition that inactivate or remove part of the *csd* allele on one chromosome [7].

All these explanations of gynandromorph phenotypes in bees imply a genetic, and thus an early, cause of the gynandromorphism. However, in the honeybee at least, male vs. female tissue differentiation develops later, at the level of cells or cell clones, according to the locally expressed form of the alternatively spliced *fem* gene, as explained above.

Here we suggest the possibility that most *Megachile* gynandromorphs are *epigenetic*, rather than *genetic*, *sex mosaics* (cfr. [58] in *Drosophila*). The adult gynandromorph pattern could derive by defects in the way the sex signal is maintained and carried on during embryonic and larval development. 

While acknowledging that the observed mix of male and female features may have been produced by either genetic or epigenetic defects, or both, because everything in the sex-determination pathway converges in the expression of the male or female isoform of the Dsx polypeptide, we think that the recurrence of the observed general pattern across so many cases in different species provides more support to an epigenetic (more developmentally linked) rather than to a genetic (topographically more random) explanation. The case of *M maritima* is so deviant with respect to all other observed *Megachile* gynandromorphs that it is not unparsimonious to conjecture a completely different (e.g., genetic) cause. In the absence of additional information, we will not discuss this single case further.

Assuming that the sex-determination mechanism in *Megachile* is the same as that in the honeybee (single-locus CSD), without appealing to haploid/diploid genetic mosaics or to homozygous/heterozygous *csd* mosaics, it derives that: (i) *Megachile* transverse gynandromorphs are diploid, otherwise there would be no mix of female and male parts; (ii) gynandromorphs are abnormal putative genetic females, because two different alleles at the *csd* locus must be present in order to have at least some body parts with female phenotype; (iii) in some tissues, those that express a male state, for some reasons, something wrong happened in the production or transmission of the feminizer signal (recall that male is the default sex); and (iv) because the female sex determination pathway is induced by the *csd* gene in early embryogenesis exclusively, a later cell lineage-specific male sex determination can derive from defects in the maintenance of the inductive signal of the *fem* gene, which fails to mediate its own synthesis.

Under the interpretation of *Megachile* gynandromorphs as epigenetic sex mosaics, and in consideration of the prevalent general pattern, the characters in the anterior of the body and in the appendices, possibly plus the scopa, are those that are developmentally (sexually) more unstable, whereas those in the posterior set are developmentally more stable. Asn present intermediate instability. Late developmental events characterized by instability of *fem* positive feedback could be associated with cell proliferation accompanying metamorphosis. The boundaries between parts with a male phenotype and parts with a female phenotype could be boundaries between compartments, or groups of compartments [59], i.e., areas corresponding to epigenetically homogeneous polyclones.

Interestingly, similar phenotypes have been also found in diploid “pseudomales” obtained in the lab by RNAi-induced knockdown of *fem* activity. *fem* siRNA was injected in honeybee embryos at the syncytial stage (0–4 h after egg deposition) and development was recorded until late pupal stage, at which the sex-specific traits of adult male vs. female can be easily distinguished [48]. Individuals subsequently determined to be diploid and genetically female developed with a mix of male and female traits, closely comparable to bee gynandromorphs found in nature, including the *Megachile* specimens described above. The most frequent condition of the hind legs among these individuals showed tibia and the first tarsus of a male-like shape, including lack of female-specific structures (pollen basket, pollen comb, and pollen brush). One individual with tibial pollen comb lacked the female-specific lobe on the first tarsus.

The interpretation of our gynandromorphs, or most of them, as epigenetic sex mosaics does not necessarily apply to other cases in Hymenoptera, including *Megachile* species. For instance, an epigenetic cause is difficult to envisage for the rare *Megachile* bilateral gynandromorphs we mentioned in the Introduction [16,17,18,19,20], because no bilateral asymmetry is expected in cell proliferation and/or in the timing of sex differentiation between the two halves of the body. 

## 5. Conclusions

Based on frequency analysis of transverse gynandromorph patterns recorded thus far in *Megachile* bees (including a set of specimens of *M. pilidens* described in this paper), the current knowledge on sex differentiation in the honey bee, and the results of recent gene-knockdown experiments in these insects [46], we suggest that these composite phenotypes are possibly epigenetic, rather than genetic, sex mosaics. Under this interpretation, the observed combination of male and female traits should derive from key developmental steps during larval or pupal development.

This is not the first example of defective phenotypes found in field-collected arthropods matching with defects obtained by manipulating the genetic control of development in an experimental setting. Uncertainties regarding the actual strictness of the equivalence between naturally occurring and experimental phenotypes are reasonable, due to the uncontrolled conditions in which the field-collected specimens developed and, also, to some extent, because of their occurrence in species other than the currently studied model species and, in most instances, only distantly related to any of them. Nevertheless, occasionally recorded specimens with well-characterized defects may offer interesting and even novel deviant phenotypes, a causal explanation of which can be at least tentatively offered. Eventually, their study may translate into the starting point for new experimental work, or a welcome test of an experiment-based hypothesis, the support for which is still less than satisfactory. In recent years, teratological arthropod specimens obtained from field collection have found a place, e.g., in the study of homeotic transformations of segmental identity [60], evolution of appendages [61], segmentation mechanisms [30,62,63], and differentiation of sensory organs [64].

Despite being generally considered marginal observations and of problematic interpretation, gynandromorphs can be an additional and even important source of data that can be considered in connection with other developmental studies across a range of taxa [65]. A methodology initiated with the study of the genetic mosaics of *Drosophila simulans* proved to be critically important in the identification of compartments [66,67], and the analysis of bilateral gynandromorphs of *Pheidole* ants revealed the presence of developmental modules [65]. In line with those investigations, the naturally occurring transverse *Megachile* gynandromorphs discussed in the present paper may provide new hints for the study of modularity and developmental stability in these insects, with particular regard to the study of the larval and pupal stages.

## Figures and Tables

**Figure 1 insects-12-00437-f001:**
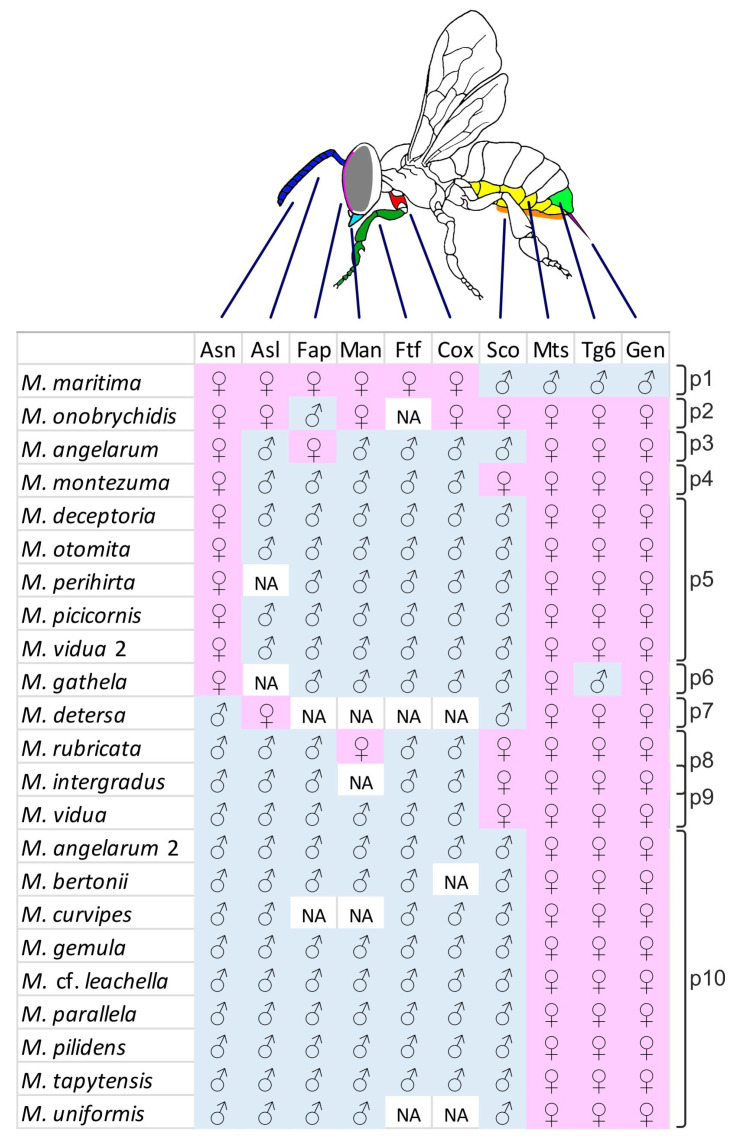
Gynandromorph patterns (p1–p10) observed in 21 *Megachile* species. Two different patterns have been recorded in *M. angelarum* (patterns 3 and 10) and *M. vidua* (patterns 5 and 9). Records with incomplete pattern description are assigned to a compatible pattern, when present. One species (*M. intergradus*) could equally be assigned to pattern 8 or 9. NA missing information. Asn, number of antennal segments; Asl, length of antennal segments; Fap, face pubescence; Man, mandibles; Ftf, font tarsi and/or femora; Cox, front coxae; Sco, scopa; Mts, metasoma segments; Tg6, sixth dorsal tergite; Gen, genitalia.

**Figure 2 insects-12-00437-f002:**
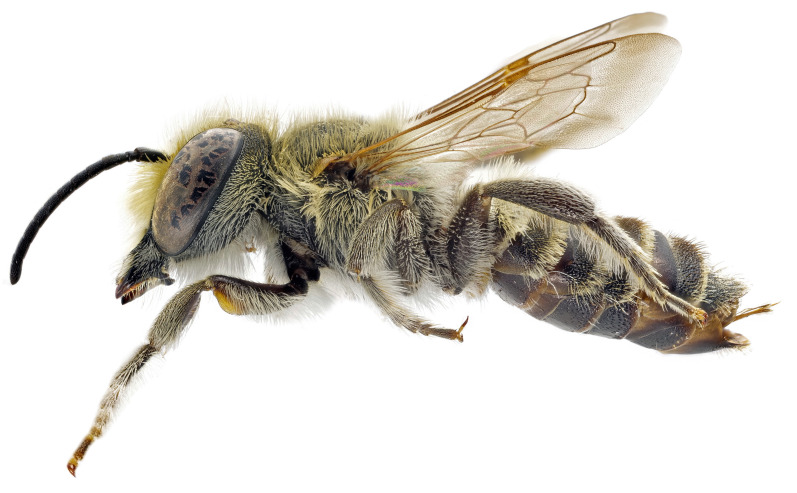
*Megachile pilidens* transverse gynandromorph from Berici Hills (North-Eastern Italy).

**Figure 3 insects-12-00437-f003:**
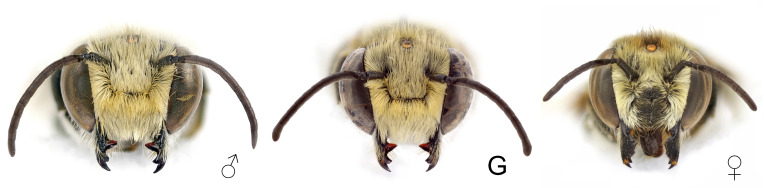
Head of *Megachile pilidens*, frontal view: ♂, male; G, gynandromorph; ♀, female.

**Figure 4 insects-12-00437-f004:**
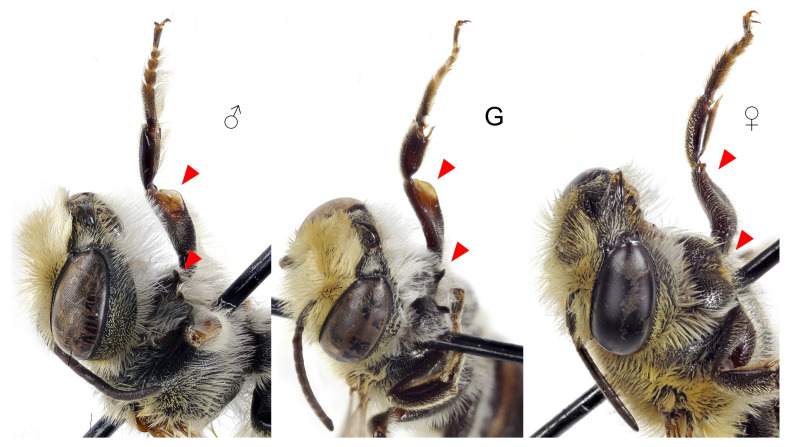
Front legs of *Megachile pilidens*: ♂, male; G, gynandromorph; ♀, female. Arrowheads indicate coxal spines and femoral dilatation.

**Figure 5 insects-12-00437-f005:**
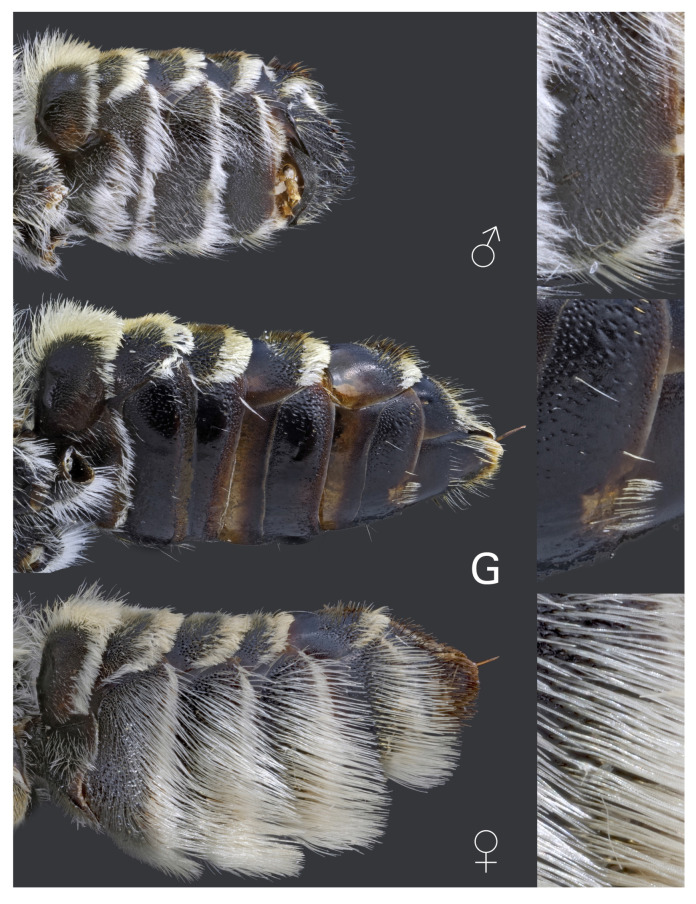
Metasoma of *Megachile pilidens* with detail of the penultimate sternite, ventro-lateral view: ♂, male; G, gynandromorph; ♀, female.

**Figure 6 insects-12-00437-f006:**
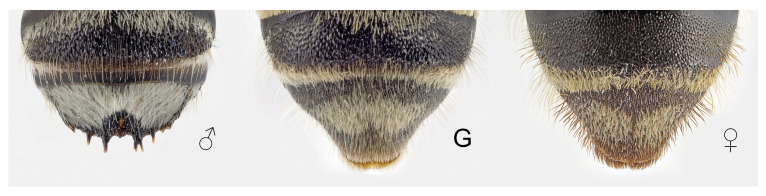
Last tergite of *Megachile pilidens*, dorsal view: ♂, male; G, gynandromorph; ♀, female.

**Figure 7 insects-12-00437-f007:**
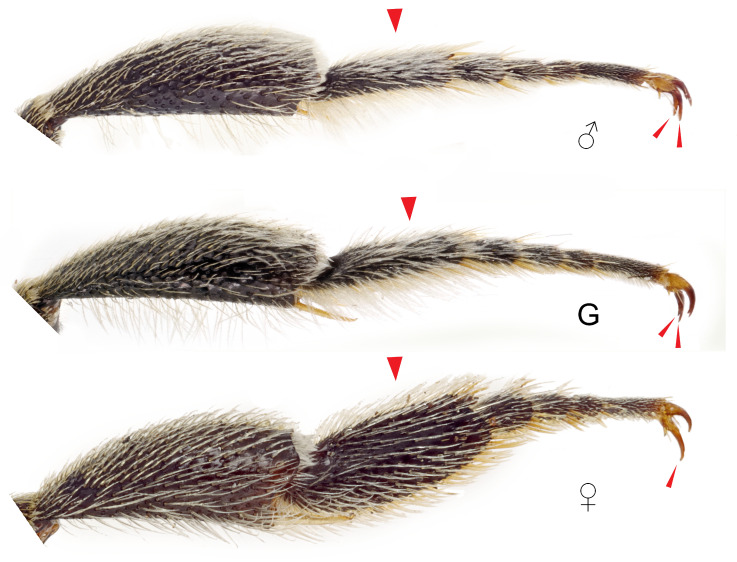
Hind legs of *Megachile pilidens*, lateral view: ♂, male; G, gynandromorph; ♀, female. Arrowheads indicates differences in morphology and setation of the first metatarsus and in the shape of the claws.

**Figure 8 insects-12-00437-f008:**
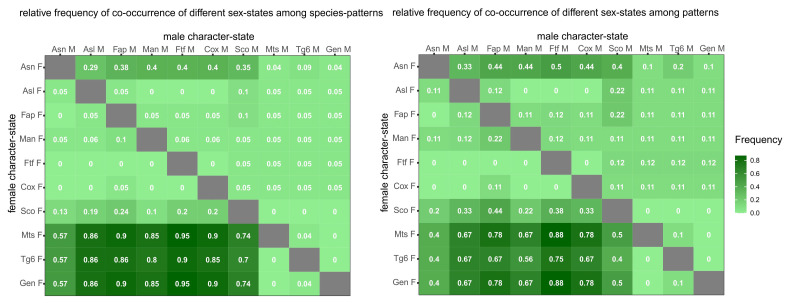
Relative frequency of pairwise combinations of male and female state for ten morphological characters. **Left**, species-pattern frequency distribution (sample size varies between 18 and 23, because some species patterns do not cover all the characters, see Figure 1). **Right**, pattern frequency distribution (sample size varies between 7 and 10, because some patterns do not cover all the characters, see Figure 1).

**Figure 9 insects-12-00437-f009:**
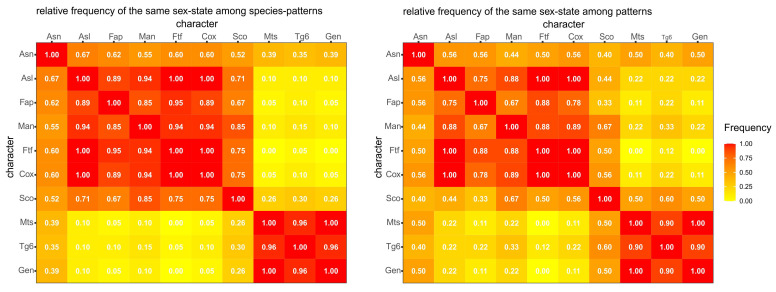
Relative frequency of pairwise sex-state concordance for ten morphological characters (the matrix is symmetric with respect to the main diagonal). **Left**, species-pattern frequency (sample size varies between 18 and 23, because some species patterns do not cover all the characters; see Figure 1). **Right**, pattern frequency (sample size varies between 7 and 10, because some patterns do not cover all the characters; see Figure 1).

## Data Availability

All data are in the article and the supplementary materials.

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
