# Peer review of "Possible Epigenetic Origin of a Recurrent Gynandromorph Pattern in Megachile Wild Bees"

_insects, 2021, doi:10.3390/insects12050437_

Round 1

Reviewer 1 Report

The manuscript is interesting. In the introduction the gynandromorph problem is correctly described and the references are enough.

In the material and methods the ginandromorph characters are clearly listed and discussed.

In the results the ginandromorph patterns are showed, and in discussion these findings are widely treated.

the images and tables are more than enough to define the biological problem of ginandromorphs in Megachilide species.

Overall, the manuscript can surely be accepted for the pubblication.

Author Response

We are grateful to rev. 1 for his/her positive comments. We have nothing to reply.

Reviewer 2 Report

Review of Sommaggio et al, “possible epigenetic origin of a recurrent gynandromorph…”

The authors present a novel (to me) hypothesis that gynandromorphs result from a mosaic of epigenetic control of gene expression, rather than a mosaic of genetically male and female tissues. They present an analysis of Megachile sp. gynandromorphs showing patterns in gynandromorph expression, and describe how what we know from honeybee genetics suggests that these are consistent with epigenetic regulation.

I have no substantial comments: I found the argument interesting, and I do not know the genetics literature well enough to spot any flaws.

My only suggestion for improvement would be the following: The authors comment that there are many reports of gynandromorphism in apis. Are these consistent with an epigenetic origin of gynandromorphy?  Likewise, are the scattered reports in other species of bees consistent?  I realize that answering my questions may require an in-depth analysis that is simply beyond the scope of the current study. However, if it is possible to comment on the fit of the epigenetic hypothesis to other empirical data, that would be a wonderful addition to the paper.

Line 36: Gonochronic- I had to look this word up, so presumably other people will too- either define it, or describe in plain language.

344: ‘to the exclusion of…” is confusing. Say something like “With the exception of…”

417: “this brought to suggest” is confusing. Say something like “This suggests…”

Author Response

We are grateful to rev. 2 for his/her positive comments. Here are our replies to his/her suggestions

The authors comment that there are many reports of gynandromorphism in apis. Are these consistent with an epigenetic origin of gynandromorphy? Likewise, are the scattered reports in other species of bees consistent?  I realize that answering my questions may require an in-depth analysis that is simply beyond the scope of the current study. However, if it is possible to comment on the fit of the epigenetic hypothesis to other empirical data, that would be a wonderful addition to the paper.

- We added a short comment at the end of section 4.3.3. (lines 482-487)

Line 36: Gonochronic- I had to look this word up, so presumably other people will too- either define it, or describe in plain language.

- Replaced with “separate-sex”

344: ‘to the exclusion of…” is confusing. Say something like “With the exception of…”

- Replaced as suggested

417: “this brought to suggest” is confusing. Say something like “This suggests…”

- Replaced as suggested

Reviewer 3 Report

This paper synthesizes earlier descriptions of gynandropmorphs in Megachile to reveal common patterns by which male- and female-typical characters are distributed along the body plan. This provides the foundation for hypotheses regarding the developmental origins of gynandropmorphs. The authors make a compelling case for their hypothesis that these alternative phenotypes arise via epigenetic processes. What makes the paper particularly valuable is that they provide testable predictions of the hypothesis.

Overall, I think paper will make a thoughtful contribution to the literature that will likely spur future research. I have only a couple suggestions for improvement. First, potential explanation for why asn and sco traits are not as reliably male or female is given briefly, but it would be nice if the paper expanded on this a bit. Especially if potential explanations could potentially provide additional testable predictions of the hypothesis. Likewise, it would be interesting to think about how the exceptions of M. maritima or the few species with bilateral and mosaic gynandromorph patterns fit within the epigenetic development hypothesis. Finally, it would be interesting to consider what is expected of the internal anatomy.

Author Response

We are grateful to rev. 3 for his/her positive comments. Here are our replies to his/her suggestions

First, potential explanation for why asn and sco traits are not as reliably male or female is given briefly, but it would be nice if the paper expanded on this a bit. Especially if potential explanations could potentially provide additional testable predictions of the hypothesis.

- We have added a new paragraph at the end of section 4.2 (lines 349-355)

Likewise, it would be interesting to think about how the exceptions of M. maritima or the few species with bilateral and mosaic gynandromorph patterns fit within the epigenetic development hypothesis.

- For M. maritima, there is nothing we think we can add to what we have already written in section 4.3.3 (lines 446-449)

- For the bilateral and mosaic gynandromorphs, we have rephrased the sentence where we cite them in the Introduction, which is now more detailed, (lines 80-83) and added a short comment at the end of section 4.3.3. (lines 482-487)

Finally, it would be interesting to consider what is expected of the internal anatomy.

- We prefer to refrain from speculating on this point, but we have added in section 4.2 (lines 332-334) a short sentence to mention the question